# Engaging with Artificial Intelligence (AI) with a Bottom-Up Approach for the Purpose of Sustainability: Victorian Farmers Market Association, Melbourne Australia

Stéphanie Camaréna [1,2]

1   School of Design, RMIT University, 124 La Trobe Street, Melbourne, VIC 3000, Australia;
    Stephanie.Camarena@student.rmit.edu.au
2   Food Agility CRC Ltd., 81 Broadway, Ultimo, NSW 2007, Australia

**Abstract:** Artificial intelligence (AI) is impacting all aspects of food systems, including production, food processing, distribution, and consumption. AI, if implemented ethically for sustainability, can enhance biodiversity, conserve water and energy resources, provide land-related services, power smart cities, and help mitigate climate change. However, there are significant issues in using AI to transition to sustainable food systems. AI's own carbon footprint could cancel out any sustainability benefits that it creates. Additionally, the technology could further entrench inequalities between and within countries, and bias against minorities or less powerful groups. This paper draws on findings from a study of the Victorian Farmers' Markets Association (VFMA) that investigated the complexity of designing AI tools to enhance sustainability and resilience for the benefit of the organisation and its members. Codesign workshops, both synchronous and asynchronous, semi-structured interviews, and design innovation methods led the VFMA to experiment with an AI tool to link sustainable soil practices, nutrient rich produce, and human health. The analysis shows that the codesign process and an agile approach created a co-learning environment where sustainability and ethical questions could be considered iteratively within transdisciplinary engagement. The bottom-up approach developed through this study supports organisations who want to engage with AI while reinforcing fairness, transparency, and sustainability.

**Keywords:** codesign; sustainable food systems; artificial intelligence; design ethics; systems thinking; farmers' markets

## 1. Introduction

Proponents of the use of artificial intelligence (AI) in food systems predict that AI-enabled technologies could see environmental benefits increase crop yields by 30%, reduce water consumption by over 300 billion litres, and reduce oil usage by 25 million barrels [1]. Overall, AI could contribute to achieving the 17 Sustainable Development Goals (SDGs) set by the United Nations [2] by enabling the accomplishment of the 134 targets under each of the goals [3]. AI is expected to not only enable the reduction of our consumption of natural resources, but to also have a more prominent role in environmental governance [4]. However, the technology comes at a significant cost to the environment due to its own carbon footprint [5,6] and the speed at which its electricity consumption is increasing [7]. Additionally, there are challenges to studying AI for sustainability due to inadequate measures of the performance of intervention strategies, increased cybersecurity risks, uncertain human responses to AI-based interventions, and reliance on historical data on machine learning [4]. Furthermore, AI is designed for technologically advanced environments, potentially exacerbating problems in less wealthy nations and increasing inequalities both between and within countries [3] and leading to problematic instances of coloniality [8]. Questions are raised both by academia and in the public discourse that were not considered with previous technologies largely because AI feels like it is more

than a tool; its utility evolves over time [9] as AI autonomy and self-learning capabilities continue to increase, making AI systems capable of learning, altering their performance, and making decisions [10].

The challenge is, therefore, to find a pathway for transitioning to sustainable food systems that leverage AI as a tool whilst still ensuring that fairness, transparency, security, and ethics are central to the AI's design and implementation. Understanding the potential catastrophes that can be enabled by failures in AI systems will require more than research based on limited datasets or prototypes [3,11,12]. AI for sustainability must be approached from a socio-technical point of view, using multilevel perspectives and systems thinking guided by robust design thinking [13]. Calls are being made for businesses working with AI to take responsibility for its use and to "hold the human at the center of all design" [14] (p. 3) and to ensure context-sensitive implementations [15]. Frameworks, tools, and checklists are some attempts at a technical fix to ethical issues raised in high-level AI ethics principles, but tangible action needs to move from high-level arguments to practice-based accountability and ethics mechanisms [16]. Failing to engage with AI in a collaborative, sustainability-driven and ethical way will open deeper issues than those made visible by AI-intensive social media platforms [17–21]. These AI-powered solutions are more than tools; they deeply influence behaviours and are pervasive. The way in which they are designed (i.e., being extremely effective at harvesting data for commercial gain) makes them extraordinarily efficient and difficult to stop. The consequences of applying AI-powered tools across food systems without thinking about their long-term design impacts could be severe.

Farmers' markets offer a lower risk and cost environment for implementing AI solutions, as well as direct access to consumers. The markets can, therefore, be effectively used to trial and test new products and learn and manage change, branding, and packaging ([22,23], p. 56 and p. 7). In our view, they represent perfect experimental environments for applications of human-driven AI in alternative food systems, a model that is easily replicated and globally applicable. Additionally, farmers' markets represent mostly small to medium farms and locally connected and biodiverse food systems. This offers opportunities to consider if and how AI could be used in the transition to sustainable systems, which differs from the current efforts of AI targeted towards 'Big Ag' or focused on AI for sorting food, food industry supply optimisation, ensuring hygiene standards, and automated food and drink preparation [24]. Considering that farms under 2ha globally produce 28–31% of the world's total crop production and 30–34% of food supply on just 24% of the world's gross agricultural area [25], engaging smaller farms in regard to AI is an important part of ensuring they are not left behind.

This article presents a case study with the aim to generate practice-based insights on engaging with artificial intelligence for the purposes of sustainability. In collaboration with the Victorian Farmers' Markets Association (VFMA) in Melbourne, Australia, the "AI for VFMA" project was conducted between December 2019 and February 2021 as an innovation process to rapid test AI solutions in the VFMA community to support sustainable food systems outcomes. Working with the VFMA offered several opportunities: (1) to run a design-led project around AI from the ground up; (2) to establish a target solely focused on sustainability supported initiatives; (3) to codesign with small-holder farmers who represent the type of communities who are left behind by most AI initiatives; and (4) to test the initiative in a rapid prototype using selected AI tools.

The case study does not focus on the findings and possible solutions generated by the collaboration with VFMA. Rather, it focuses on the iterative design research process itself. To the best of our knowledge, no study has yet been done on ways of considering AI in alternative food communities to support and scale-up their sustainability efforts. The case study looks to answer the following questions:

- Can the design process support the selection of AI tools for sustainable food systems?
  - How does the design process carry the sustainability intent?
  - Where are questions of AI ethics discovered and resolved?

○ Can the process avoid technological solutionism? If so, how?

- Can the process be reproduced to support transition to Sustainable Food Systems (SFS) using AI?

Design guiding principles are provided in the following subsections to set the context of the research at the Victorian Farmers' Markets Association (Section 1.1), explain the ethical (Section 1.2) and sustainable (Section 1.3) boundaries the study it is situated within, and the engagement model of codesign (Section 1.4). Two areas of focus were adopted: system thinking as an epistemological lens [26,27] (pp. 139–158), and collaborative transdisciplinary processes [28,29] codesign [30]. These were adopted because they are particularly suited to sustainability research as a complex and wicked problem.

### 1.1. Victorian Farmers' Markets Association

Farmers' markets provide an avenue for food producers to sell produce from their farms directly to consumers. In Australia, this includes primary food products, seafood, game and foraged foods, value-added foods, specialty food products, regional produce, garden inputs, and small livestock. Non-farm products, craft, pet food, and books are not recommended for sale at farmers' markets in Australia [31]. Colmar Brunton [32] reported that 19% of Australians source their fresh vegetables from farmers' markets and that 4% purchase them directly from the producer. For a quarter of farmers' markets stallholders, 75% of their produce is solely sold through the markets, while the vast majority use a range of other distribution channels [22] (p. xii). There are 193 farmers' markets in Australia registered with the Australian Farmers' Markets Association [31], 69 of these are in Victoria, of which 33 are accredited by the VFMA.

The VFMA is a values-driven organisation designed to support Victorian growers to sell their produce directly to the public. The Association is backed by the farmers' markets accreditation model, which renders it unique in Australia and provides a guarantee of provenance to the public. The VFMA was founded in 2004 as a not-for-profit membership association. In 2011, the association created and registered an accreditation programme with the Australian Competition and Consumer Commission (ACCC). The programme certifies authentic farmers' markets for producers and consumers and ensures that the majority of produce sold at accredited farmers' markets have been made or grown by accredited producers [33]. The strategic intent of the Association focuses on sustainability for both the growers and the Association, the provision of healthy, locally grown food to the communities it serves, and the aim to build food resilience across communities of growers and Victorians [34] (p. 3). In 2020, the Association membership was composed of 33.9% accredited fruit/vegetable producers, 26.6% non-accredited market traders, 21.1% accredited specialty makers, and 10.7% accredited meat/dairy/eggs producers [34] (p. 4), representing over 600 members.

Engaging with AI was not a natural next step for the VFMA, but it is representative of the mindset displayed by the Association's committee, which is to remain open to innovation and the creation of knowledge to serve its members and the public. It is also representative of the farmers' markets in general who offer a perfect vessel for service and product innovation with direct access to both growers and customers and the ability to quickly test ideas in the field with fast feedback [22] (p. 13). Testing AI in the context of an Australian farmers' market association and in collaboration with small holder farmers allows a focus on the design and development of AI solutions in areas with localised problems, albeit in a wealthy nation.

### 1.2. AI and Ethics

The field of AI is wide and diverse and includes sensing, modelling, planning, and action, as well as decision-support systems, natural language processing, perception, analytics, and robotics [35]. In agriculture, AI-powered tools help farmers test soil [36], identify crop disease [37], determine food quality [38,39], quantify and predict crop yield [40], and take a whole-systems view of agricultural data [41]. While the wider research project on

which this case study is based was initially set up to include any applications of AI, it centres on simpler and more accessible AI-tools to take into account both the participants' knowledge and understanding and the project's budget and timelines. Therefore, when referencing 'AI' in this paper, we refer to AI-powered tools (for example: tools used for the detection of pests and diseases, or to count fruit), rather than complex autonomous systems like robotics or autonomous machinery.

AI ethics is a growing field. When considering the ethics of AI systems as objects (or tools made and used by humans), issues emerge primarily in relation to privacy, opacity, bias, human-machine interaction, employment, and the effects of autonomy. Ethics concerns also apply to AI systems as subjects (ethics for the AI systems themselves) in the fields of machine ethics and artificial moral agency. This paper focusses on AI systems as objects or tools [42].

Several principles and guidelines for ethical uses of AI have emerged over the past few years and converge around five ethical principles: transparency, justice and fairness, non-maleficence, responsibility, and privacy [43]. However, most of these guidelines remain high-level, and in practice have little to no enforcement mechanisms [44] and little regulation [45]. Many of these guidelines will be of little influence if they continue to follow a top-down approach [14,46,47]. Most codes are self-regulatory and difficult to integrate into existing sector-specific law and policy, largely because AI operates across sectors [47] and across national boundaries [48]. Governments often discuss protecting the human workforce, but there is no clear policy vision [10,49]. Many experts are particularly calling for the protection of constitutional democracy and want to see a clear demarcation between what can be handled by AI ethics principles and what should be addressed by laws [45,50–52].

To provide guidance on the selection and use of AI tools that the case study uncovered, we selected the AI Ethics Principles framework developed by the Australian Government's Department of Industry, Science, Energy and Resources [53]. The framework was adopted after extensive consultation with businesses, academia, and the community, and aligns with globally agreed values. The framework provides the prompts necessary to bring the high-level principles into practice, an important step for the implementation of AI ethics principles in a bottom-up approach.

*1.3. Sustainable Food Systems Framework and Leverage Points*

A systems thinking approach to sustainable food systems considers the interconnections and systemic impacts of transition actions [54]. The ability to zoom in and out provides a viewpoint required to coordinate transition activities, which transcend the traditional boundaries of a food system [55,56]. Food systems are complex, hierarchically nested and adaptive systems, which interact with each other and show adaptive and emergent qualities [57]. Managing this complexity requires the use of systems thinking to work on multiple leverage points (influences) to build on the emerging new paradigms and to be able to act, learn, and plan at the same time [58].

Leverage points are places to intervene in a system [59]. A systems approach to leverage points is relevant in the transition to sustainable food systems and the achievement of the UN Sustainable Development Goals [26,60,61]. For Abson et al. [26], the twelve leverage points proposed by Meadows [59] relate to a hierarchy of four system perspectives on sustainability [62]: parameters, feedback, design, and intent. These characteristics influence a system at a shallow (parameters, feedback) or a deep (design, intent) level and, therefore, respectively display a lower to greater ability to bring transformational change to the system they operate within; the type of change needed for sustainability transformation.

In the narrowed context of farmers' markets, small producers, and local food systems, this case study used the Intergovernmental Panel for Climate Change's (IPCC) food security framework [63] to scope the problem definition and solution development for the purposes of sustainability (Figure 1).

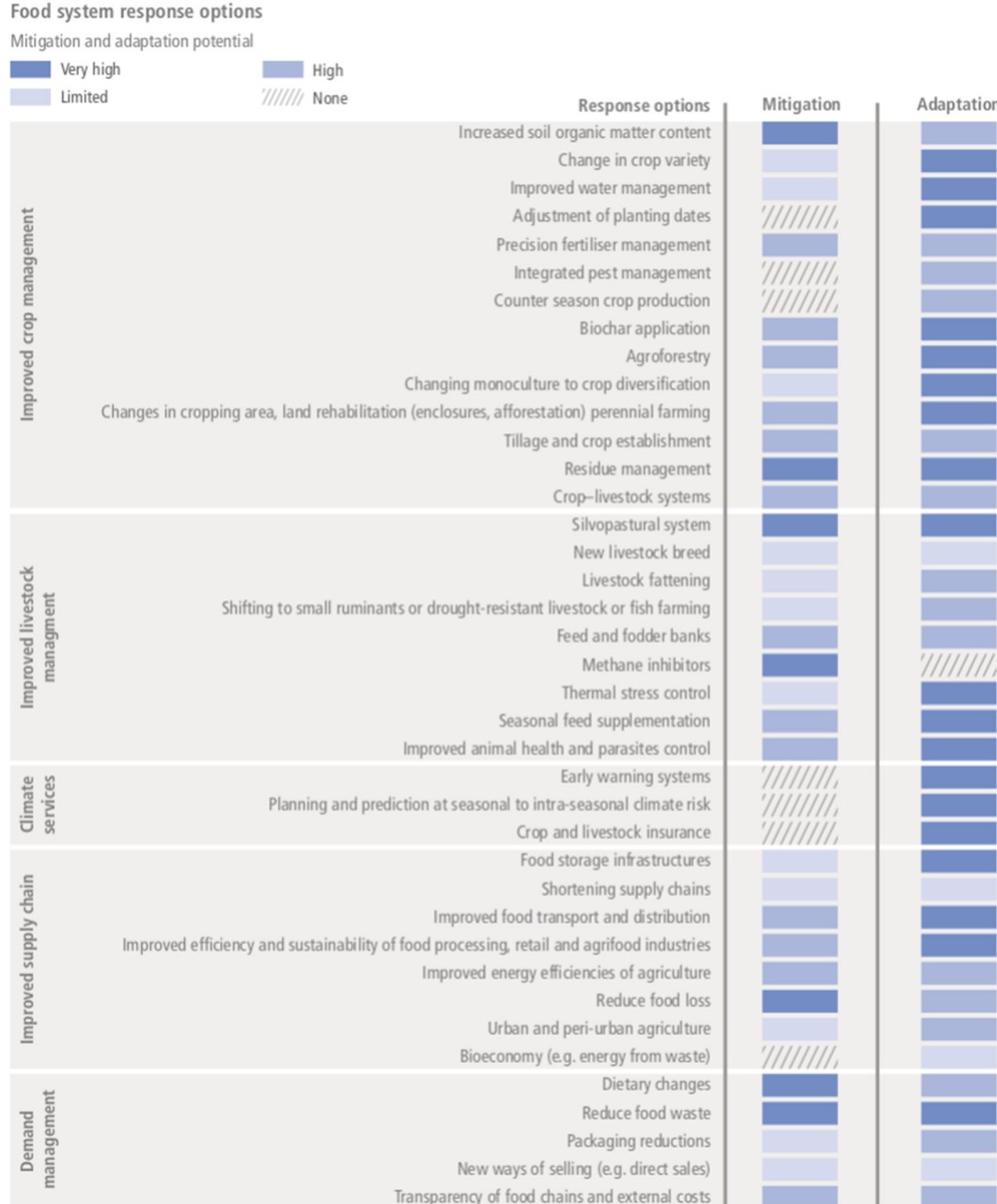

**Figure 1.** Selected SFS framework extracted from the IPCC climate change and land special report [63] (p. 493).

The IPCC's framework mapped more closely to the activities of the VFMA and the member producers. It provided methods for measurements, which, while not used in the case study, leave room for intricate improvements of the wider project's reporting over time.

*1.4. Codesign*

Codesign refers to collaborative creativity across the design process, which includes both designers and people not trained in design, working together [64]. The collaborative design framework used in this research project [65] underpins the designer's style of guidance used in the codesign activities. The guidance of the designer varies between facilitation and steering at various stages to discover and explore options, imagine and consider options beyond the world as it is, expand and consolidate options, and create, envision, and develop options. Codesign, in the context of this article, is expressed both

in terms of formalised processes of creating a common language and shared visions and strategies, but also as a social conversation in which different actors interact in different ways and at different times [66] (p. 49).

In Section 2, we introduce the methodology of this study, and in Section 3 present an analysis of the codesign activities results. Section 4 discusses the insights on the research questions, proposes an approach to the use of AI in sustainable food systems and suggests further research.

## 2. Methodology

The case study was initially planned to run over a period of between three and six months, but this was extended to seven months due to the COVID-19 pandemic that impacted the sites between February and September in 2020. The general approach to the project was transdisciplinary and participatory. This provided the guiding principles to engage with AI within a complex system environment, with humans at the centre and informed by multilevel perspectives. The exploratory case study built gradual common knowledge about the subject and its relation to participants in a bottom-up driven process. This was done by selecting three groups of participants from different disciplines and backgrounds who would inform and guide the process (Section 2.1). These groups are a crucial element when working with AI [13] to ensure that a diversity of at times opposing views are represented in the analysis [28]. Iterative and codesign approaches supported the problem definition and solution development (Section 2.2). The transdisciplinary co-creation framework [67] (p. 191) is useful in supporting the development of balanced AI scenarios (across social, technical, and environmental realms) [45,68]. A sustainability and ethics validation framework anchored intent and goals of the VFMA's project (Section 2.3). The viewpoint from which we conducted the case study—design and systems thinking—allowed us to see solutions within a whole-system context, which then allowed us to better identify the social and environmental costs of AI and their trade-offs with the benefits of AI [4,13,26]. The results were then analysed to reveal the elements in the case study, which could be reproduced for similar research (Section 2.4).

The methodology for this case study was derived from previous research [68] investigating how AI can support the design of transitions to sustainable food systems [4,13,68]. That research found that designers need to consider sustainability as the main intent inside an ethical framework, otherwise the use of AI in food systems would be disconnected from sustainability outcomes. As a result, the designer's intent, using the foundation of sustainable design strategies, becomes the main driver of sustainability throughout the problem definition, solution development, and implementation phases of an AI for food sustainability project. The highly participative approach meant that activities often emerged from previous activities.

### 2.1. Participant Engagement: Who Sits at the Table

In the words of Costanza-Chock [69] (p. 84), "Don't start by building a new table; start by coming to the table". An important part of codesign activities, Costanza-Chock argues, should be a willingness to bring design skills to community-defined projects, rather than seeking community buy-in to externally defined projects. This case study did not have a defined project that then sought community buy-in, but rather invited community participants to develop their own project regarding the use of AI at VFMA markets.

Three groups of participants were involved throughout the VFMA project (Figure 2). The first group, named the "internal panel", was composed of one or two representatives of the VFMA board and the author. The second group, the "extended panel", included the internal panel plus other VFMA direct participants such as producers, designers, and food systems professionals. The third group, the "external panel", is the only non-homogenous group, in that its members never met. The external panel was composed of academic experts in fields as diverse as nutrition and machine learning, meat standards and soil health, consumer perception and agronomy. Other participants in the third group

were industry experts in the fields of artificial intelligence, regenerative agriculture, and expert growers (such as cattle breeders and agronomists). In total, twenty-seven people participated in the project.

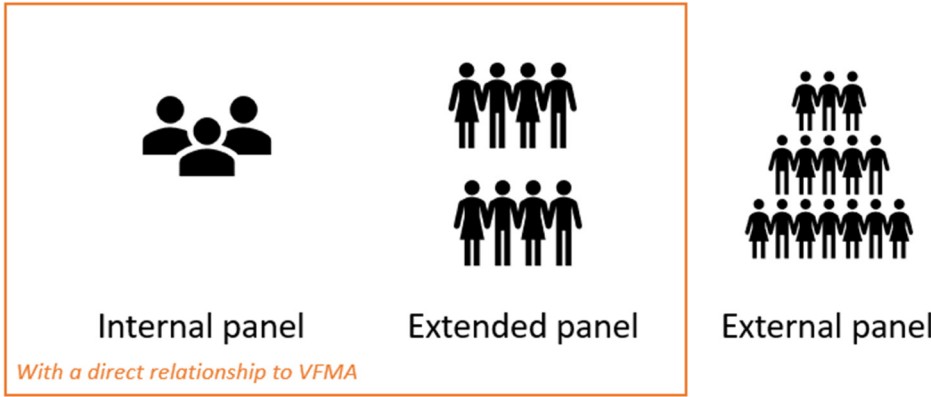

**Figure 2.** Three groups of participants involved throughout the process and collaborative activities.

Each group was called upon at various stages of the process in a series of collaborative activities. All three panels constituted a transdisciplinary platform for knowledge to circulate between.

The internal panel met once a week to discuss what was undertaken in the week prior and any new information discovered either from research actions or from input from the extended and external panel. Findings were reflected upon and informed the design of the following week's project activities. Activities were planned over a two-week period using an online tool, Trello, to manage the project activities and comments (Figure 3).

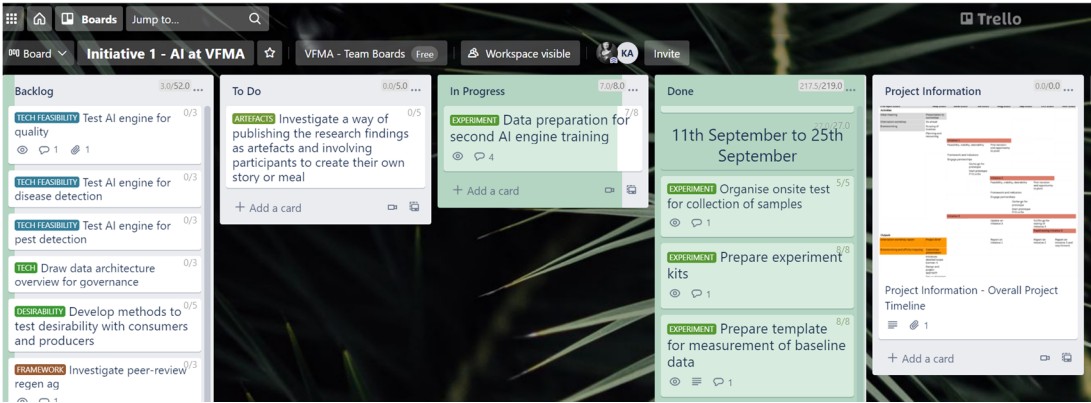

**Figure 3.** Trello board to manage and communicate about the project activities.

The extended panel was involved in all the collaborative sessions (Stages A through D) but also received and discussed a monthly summary of the research that they could provide feedback on and use to help design the next stages.

The external panel's input was available outside of the project activities, and external expert input was gathered through semi-structured and unstructured interviews using emails, and phone and video calls. Part of the iterative learning revealed new questions that required experts in the field to clarify or guide the next steps of the design process. The external panel members were referred through existing and emergent relationships and were never engaged as a group. For example, when the extended panel decided to investigate measuring the quality of meat, meat producers (VFMA members) were invited to inform the panel, as were academic experts in meat standards and nutrition.

The case study participants decided to scope the project within the boundaries of the VFMA and its members. This was related to issues of time availability, limited access to

consumers due to COVID-19 restrictions, and the need to understand for themselves what a project would look like before involving consumers and other partners. However, involving additional groups was discussed and flagged as a natural part of the next iterations. While the VFMA customers (who attend the markets) were identified as a core group in the study, they were not engaged in the case study activities due to the difficulty with engaging with the public at that time. This is an area that should be further investigated.

### 2.2. Process Activities: How We Engage

Agile iterative practices were used to create ongoing cycles of learning. Through iterative and incremental project management, the participants operated within cycles of plan-design-check-adjust, or what is known as the Deming cycle [70] (p. 88), which privileges progress over perfection. The Deming cycle also encourages validated learning, which is "not after-the-fact rationalisation or a good story to hide failure" [71] (pp. 6–7), but a rigorous method for demonstrating progress and a process of testing hypotheses with empirical experiments, to first ascertain not only if a product (or service) can be built but if it should be built in the first place. Progress was, therefore, measured in terms of participant feedback rather than milestones. Tasks were planned and delivered within two-week periods and weekly meetings allowed an iterative review of the progress, questions to be raised, and reflections to be shared. Agile iterative practices are commonly used in software development practices and closely tie in with design thinking practices [72–75].

Codesign activities engaged the different panels in a series of activities to orient the work (Stage A), develop a strategy (Stage B), scope the initiative (Stage C), scope an experiment to test the initiative (Stage D), and finally to run the experiment. An additional stage was added, Stage C+, which was initiated following a Stage C decision to further investigate the definition of 'quality' (see Figure 4). Stages A to D are discussed in this article.

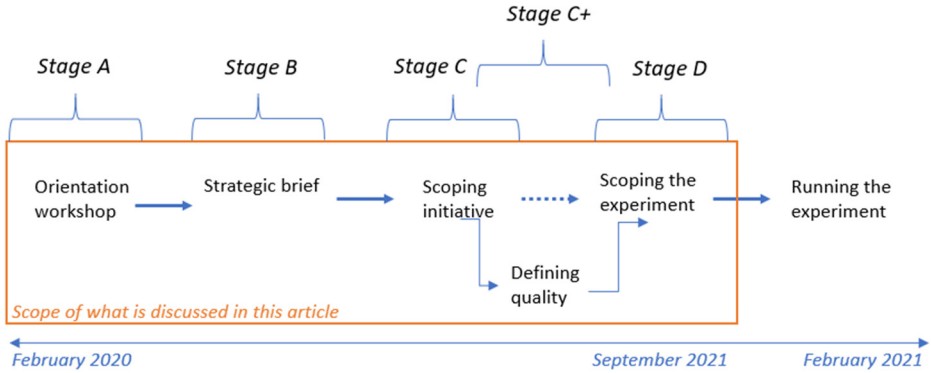

**Figure 4.** Project stages, scope, and timeline.

Cocreation approaches for transdisciplinary research are particularly suited to deal with practices where a level of uncertainty and complexity emerges with the integration of different perspectives and expertise of participants from inside and outside academia [67] (p. 191). Agile project management and codesign both evolved from user-centred co-creation practices and when integrated provide opportunities to capture interactions and feedback loops among a technology, users, and other stakeholders [4]. The engagement model needs to cater for progressive adaptation of a shared language and different types of tools [76] among the participants. This is a prerequisite when the priorities are centred on the complex areas of sustainability, food systems, and AI, a technology that requires a new type of thinking.

Codesign activities were attended by the extended panel. They began with an orientation workshop (Stage A) to explore the context within which the Association, its members, and customers operate. It aimed to draft a vision of what the Association could become and how AI tools could support that. The workshop followed a model proposed by Sniukas [77]

using an agile sprint-based approach to making and executing strategy, which is particularly suited to complex environments [78]. The neutrally facilitated conversation aimed to deliver a vision of what the group would like to achieve, independent from the tools (including AI) that could help achieve it. The conversation and notes were captured on a white board under four categories: (1) the questions that the conversation raised ('what is' and 'how is'); (2) statements made by participants that were flagged as assumptions ('I believe'/'I don't believe'); (3) opportunities to explore ('I think we should'); and (4) strategic moves ('we clearly know what and how to do it and why'). Following the workshop, brainstorming activities were organised via email when availability and time constraints were limited and in-person meetings impossible in the context of the pandemic and lockdowns. These emails acted as an asynchronous method to gather ideas and suggestions. It was difficult to align meeting availability when the participants were also small-holder farmers with little time to spend outside their business but allowing and designing asynchronous participatory design activities alleviated this issue ([65,79] p. 30 and p. 279). Affinity mapping [80] (p. 127) followed, which helped categorise the main ideas raised in the workshop and the brainstorming activities into themes.

Following the initial workshop and the identification of key directions highlighted in the themes, a strategic brief (Stage B) was prepared, which mapped three proposed initiatives against the Associations' strategic pillars and provided some estimation of what the Association could achieve by engaging with the project. These were presented to the board in an online session where they were thoroughly discussed and evaluated.

The third collaborative activity (Stage C) was organised online using Mural, a digital platform for visual collaboration. A lean canvas, one-page business plan, shown in Figure 5 [81], was used to support the session facilitation. Based on the original business canvas developed by Osterwalder [82], the lean canvas is a fast, concise, and effective tool to quickly flesh out new product and service ideas while identifying the best idea and validating it.

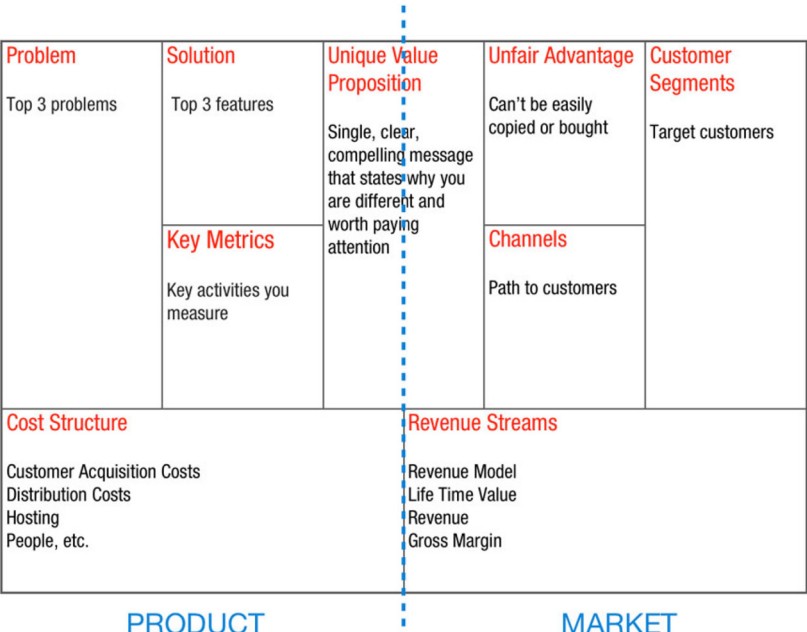

**Figure 5.** Blank lean canvas in Maurya's Running Lean: Iterate from Plan A to a plan that works [81].

The participants were guided through the Mural board via a Zoom session. The following areas of the canvas were in focus: problem, solution, customer segments, unique value proposition, cost, and revenue. A short timeslot (an hour and a half) was allocated to build on previous work, focusing on problem definition and workable solutions that the panel should consider.

The lean canvas was created with the aim to quickly identify the assumptions made by the team about each of the dimensions (problem, solution, customer segments, etc.). The riskiest assumptions were then highlighted and agreed to be tested in the field before anything was committed to the canvas. Assumptions were often about who the targeted customer is, what was the pain point attempting to be solved, or how a solution might solve a problem.

Due to the short time allocated, the identification of the riskiest assumptions had to be conducted asynchronously. The post-it notes on the canvas were mapped and sorted by affinity to get clearer avenues for action and the selection of an experiment. The riskiest assumptions were identified on the board and discussed offline with different members of the team.

At the end of Stage C, it appeared that "food quality" needed to be defined more precisely before the team could decide which problem they wanted to solve and how. Each participant had a slightly different definition for it, and this created confusion as to what the project was going to set as a success criterion. The panel decided the definition of "quality" was an important assumption to test and validate in order to avoid design problems [83]. As a result, an additional collaborative activity was created (Stage C+), which focused on defining "food quality" in the context of the case study.

Stage C+ investigated what "food quality" meant for the committee and for VFMA members and aimed to clearly define what AI could do to support or measure it. An asynchronous exercise, using an online shared document, was conducted with the extended panel. Participants were asked to list three to five adjectives they felt best characterised attributes of "quality" in the context of farmers' markets produce. They also had the opportunity to add or expand on definitions. Once the extended panel agreed on their top five adjectives, the definitions were used to elicit further input from the external panel. For example, "high nutrient density" was one of the definitions retained, therefore, one of the questions put to the extended panel was "how do you measure nutrient density?".

Stage D built on the previous stages and aimed to design an experiment where the VFMA would use AI to respond to the problem identified collaboratively. That phase aimed to both create a list of potential AI-powered solutions, and an evaluation of these solutions. The original activity planned for Stage D was an online workshop where the extended panel was asked: "how might we create an evidence-based story of quality at VFMA so that producers and consumers find value in it?" "How-might-we" questions were used to define and frame a design challenge, "opening the field for new ideas that we do not yet know the answers to" [84] (p. 125). The online collaborative session used a Mural board, and the panel was asked to develop a hypothesis using the frame: "We believe that . . . will drive . . . within . . . ". The session guided the participants towards defining the high-level steps required to test the hypothesis identifying who can help, the resources needed, and the risks. However, the session was not entirely successful, and the results obtained were limited. As a consequence, the internal panel devised a different approach, narrowing the exercise to an evaluation of the specific AI solutions discovered throughout the process against design innovation and AI ethics principles, a sustainable food systems framework, and systems thinking leverage points.

### 2.3. Evaluation Framework for AI-Powered Solutions

Throughout the codesign process, engagement with the different panels and concurrent research conducted by the author resulted in a list of seventeen AI-powered solutions, which could serve the VFMA's sustainability intent, either as they were designed or with modifications.

The solutions were evaluated in a prioritisation table against four dimensions: design innovation, AI ethics principles, sustainable food systems, and system leverage point. The design innovation dimensions determined if the AI-powered solution was a viable, desirable, or a feasible solution. In the design innovation context, feasibility allows for the understanding of technological capacity; viability focuses on understanding customer

affordability; and desirability takes into consideration the users, their social-cultural context, problems, needs, and desires [85]. The AI ethics principles presented in Section 1.2 guided the evaluation of each solution. The internal panel established whether each solution followed the AI ethics principles or could easily be modified to comply. For example, when evaluating a digital agriculture service platform, the panel found that while the platform included a number of climate change metrics, the IPCC principles required an option to add human and social metrics to the datasets if this solution was to be selected. The sustainable food systems principles established in Section 1.3, were used to reflect on the ability for a solution to support the IPCC food security mitigation and adaptation strategies. For example, some solutions would directly improve crop management, generate climate services data or improve the supply chain. Finally, intent on viewing the potential solutions within a system thinking lens, the internal panel evaluated the possibility that a solution would impact food systems in small (parameters and feedback) or significant (design or intent) ways (see Section 1.3 for further theoretical background). Each dimension was broken down into sub-dimensions where relevant; some of which referred to the VFMA's strategic pillars to ensure the overall direction of the project remained within the VFMA's main goals and those of its members (Table 1).

**Table 1.** Dimensions and sub-dimensions in the AI tool evaluation process.

| Dimension | Sub-Dimension | Focus |
|---|---|---|
| Design innovation | Desirability | Producer |
| | | Consumer in person |
| | | Consumer online |
| | | VFMA |
| | Viability | Cost/ROI for user |
| | | Cost/ROI for VFMA |
| | | Alignment to VFMA strategy |
| | Feasibility | Timeline |
| | | People/skills |
| | | Solution |
| Additional opportunity for innovation | | |
| AI ethics | Human, social, and environmental wellbeing | |
| | Human-centred values | |
| | Fairness | |
| | Privacy protection and security | |
| | Reliability and safety | |
| | Transparency and explainability | |
| | Contestability | |
| | Accountability | |
| SFS sustainability | Improved crop management | |
| | Improved livestock management | |
| | Climate services | |

**Table 1.** *Cont.*

| Dimension | Sub-Dimension | Focus |
|---|---|---|
| | Improved livestock management | |
| | Climate services | |
| | Improved supply chain | |
| | Demand management | |
| System leverage point for transition | Parameters | |
| | Design | |
| | Feedback | |
| | Intent | |

The prioritisation table was shared with the internal panel. For each solution, the author provided comments on the dimensions and sub-dimensions for the other panel members to review. Each member reviewed the content and collaboratively established an order of priority for the most appropriate solution at that time. The scoring of each solution was initiated by the author but not followed by all the members and was, therefore, not considered in the prioritisation exercise, which was organic.

*2.4. Data Collection and Analysis*

The analysis, driven by the research questions outlined in Section 1, aimed to observe the design activities and their results to understand which parts are reproducible and how. Where the evaluation framework provides an indication of how the solution furthered sustainability at the VFMA, the design activities indicate ways in which the process balanced socio-technological engineering with the environment. Analysing the process and its contribution to sustainability-supporting solutions provides a reproducible means of engaging with AI in other sustainability conscious communities.

In order to analyse the case study findings, we used the concept of "nature versus nurture" as an analogy to represent how sustainability considerations were included in the process. "Nature" was used to refer to the inherent values of the people or infrastructures which carry out a particular sustainability intent; "nurture" was used to refer to the elements in a process which, due to their design, might elicit the integration of a sustainability intent. The case study notes, interviews, and artefacts were reviewed to identify where sustainability was mentioned, or sustainable values were demonstrated, and these findings were then listed in a table categorised for each stage.

Reading, reviewing, and reflection informed the analysis of ethics in the case study, how and when it appeared, and whether it was specifically related to AI or a more generic approach to ethics. The limitations of that exercise are due to the author's focus on AI ethics without the philosophical depth an ethicist would bring. However, the evaluation remains valid because very few people trying to navigate AI ethics principles will have any type of in-depth knowledge of ethics. Therefore, the findings could help inform a broader section of the public intending to use AI in sustainable food systems.

Finally, we analysed the concept of AI from the initial stages to the selection of an experiment to understand how it evolved and what it led to. This was done by paying attention to the crafting of the design activities questions and themes in the material and the final list of AI-powered tools considered for experimentation.

## 3. Results

In this section, we reflect on the results of the codesign activities in the context of the research questions. Figure 6 outlines the relationships between the groups and their influence on the project's design and progress. It summarises the elements and timeline of

the design process, the impacts of transdisciplinary practices on divergent and convergent thinking, and instances of when AI conversations took place among the participants. The information contained in Figure 6 is used as a background for the results presented in the following sub-sections.

**VFMA - Project Timelines and Activities**

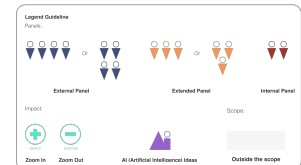

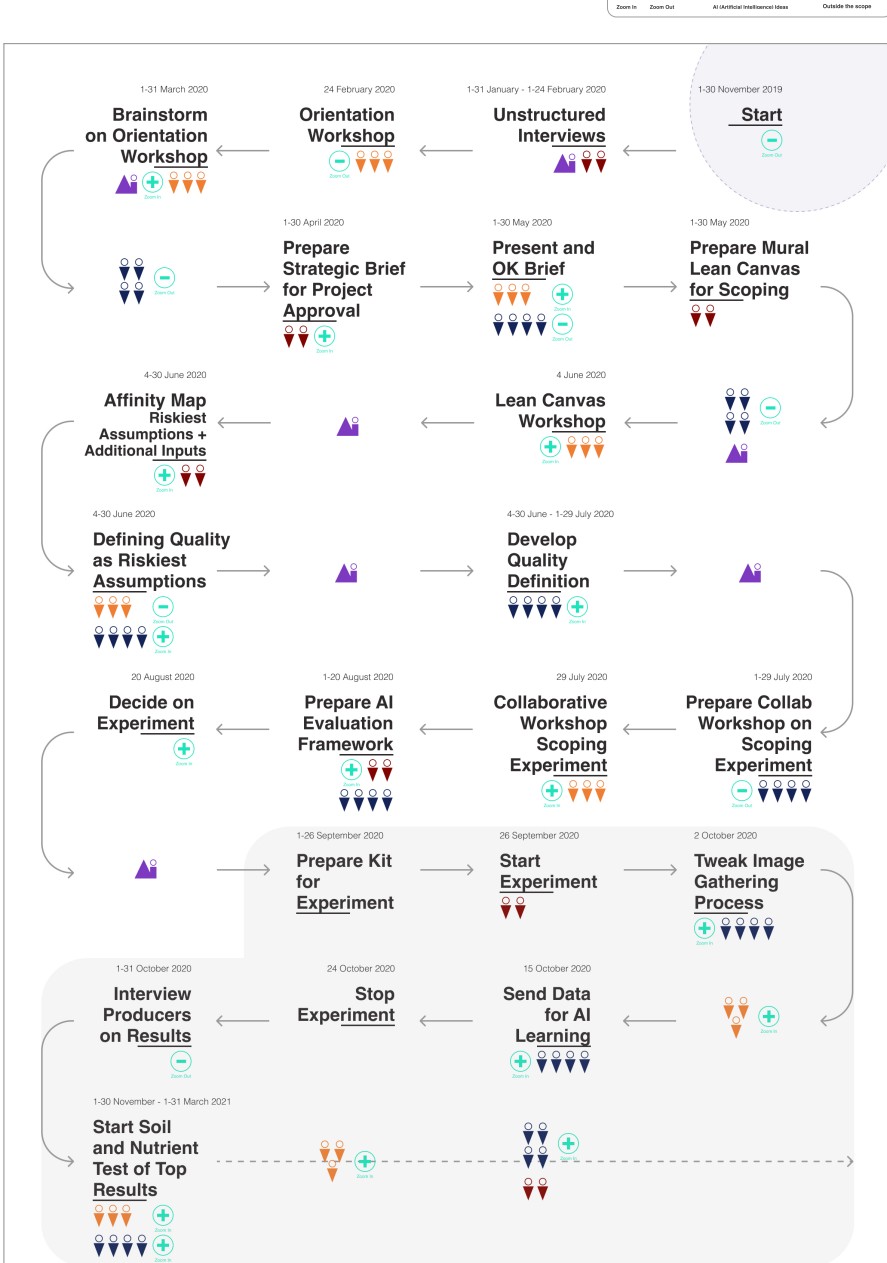

**Figure 6.** Activities, participants' intervention and types of influence, AI conversations are mapped to the project timelines.

Section 3.1 establishes the role the design process played in maintaining the sustainability intent throughout the codesign process. Section 3.2 reviews how AI ethics principles appeared and were dealt with. Section 3.3 pays attention to how the understanding of AI was reflected through the design activities and results.

### 3.1. Sustainability in the Design Process: Nature versus Nurture

Transdisciplinarity and codesign were established as the principal approach to the case study (Sections 1.3 and 1.4). The transdisciplinary platform engaged the three panels in the design activities around a specific *problematique* where co-production and integration of knowledge takes place [86]. Integrating sustainability as a shared vision into the project was completed through the design process (Figure 1) but also relied on what the participants brought to the table. Table 2 presents the way sustainability was considered, i.e., as nature or nurture, across the different design activity stages.

**Table 2.** Nature versus nurture when integrating sustainability throughout the project implementation.

| Design Activity | Sustainability Considerations | |
| --- | --- | --- |
| | **Nature** | **Nurture** |
| Stage A—Orientation workshop | Participant's personal involvement over the years with sustainability and their dedication to both food quality and solving sustainability issues. Panel genuinely care about the Association, their members, and the farmers' markets (FM) customers. Strong values on collaboration, recognition of farmers' knowledge, regenerative practices in agriculture, financial and environmental sustainability. | Theme selection: How might we use AI to scale up the VFMA members' sustainability impacts? Introduction: Participants share their motivation for being present and the type of challenges they feel are important with regards to sustainability. Intent: Establish a vision independent of the tools. |
| Stage B—Strategic brief | Strategic pillars of the association focused on the diversity of produce and accredited producers, financial sustainability for the members, feeding the FM supporting community, and building a resilient food system. Participating panel provided ideas and defined parameters for benefits of the research project. | Alignment: Proposed initiatives alignment table to VFMA strategy. Nothing is off the table: ideas are proposed but open for complete redesign. |
| Stage C—Scoping an initiative 1 | Participant self-evaluated and reviewed ideas and proposals on the board in light of the values they share. Strong ethical and sustainable values informed the conversation and the problem definition. Pragmatic approach kept ideas on the practical rather than theoretical side. | Strategy blueprint: Guide to the session to determine if the canvas findings meet the challenges, the aspirations, focus areas, guiding principles, and outcomes identified to date (see Supplemental Materials Figure S1 for more details). Visual collaboration: Setup of the Mural board allowed all participants to add their ideas but also to read others', which in turn triggered reactions and further inputs. Lean canvas: Format guided dimensions of social, environmental, and financial sustainability questions. Riskiest assumptions: Canvas facilitation towards identifying risk clarified assumptions that carry potential impact on sustainability and ethics. |

**Table 2.** *Cont.*

| Design Activity | Sustainability Considerations | |
| --- | --- | --- |
| | **Nature** | **Nurture** |
| Stage C—de-risking assumptions about food quality | Panel definition of "food quality" relates to holistic view of quality and integrates sustainability across all phases of the produce. Lived experience from producers brought the concept of "story of quality" to the project. Beginner's mind: Seeing old information with new eyes happened when the designer questioned what is taken for granted. | Openness to change: Organise a new collaborative design activity to investigate deeper link to values. Listening to different voices: A simple exercise led to a holistic view of food quality. More than the sum of the parts: Active seeking of external panel's input brought rich nuances to the definition of quality and ways to measure it. |
| Stage D—Scoping an experiment | Intimate knowledge of the VFMA and its members provided depth to the selection of a solution to scale up sustainability. The VFMA has direct relationships with producers and consumers, making experimenting with AI for sustainability simplified. VFMA is a small representation of bigger systems and a model of alternative food systems where AI for sustainability can be studied in decentralised environments for small to medium producers. | Awareness: Recognising when too much collaboration impedes the work and pivoting to narrower input was required. Prioritisation table: Provided a frame for evaluating AI solutions against innovation, sustainability, ethics, and systems. Building in iteration: All the work to date informs the evaluation exercise and builds on what was learned, therefore, decisions were based on the collaborative work but finalised by the key carrier of the project, intent. Lean startup: Creating a tangible artefact for the problem we want to solve and the solution we want to try forces the initiative to become a practical experiment which can be built upon (persevere or pivot). |

The inherent values of the internal and extended panels clearly influenced the thinking about sustainability throughout the process. This is something most panel members have been involved in for a long time or have a keen interest in. The respective panels' input corroborated these values in all collaborative sessions conducted both synchronously and asynchronously. The codesign process supported deliberate practices for transformation and encouraged new ways of thinking about problem formulation. For example, the format of the orientation workshop allowed the participants to connect to *why* they had decided to be involved in the AI for VFMA project. When envisioning the future of the VFMA and the place AI could have in that future, ideas originated from a joint vision of sustainability co-created in the session (Stage A). Joint visioning is a key method in transformational sustainability science and an influential stimulus for change. Table 3 illustrates the resulting themes mapped as a result of Stage A. The results show the holistic view of sustainability from the participants, and a first attempt at imagining AI solutions aligned with their values and included in the visions of the future of VFMA.

**Table 3.** Stage A—Orientation workshop—Themes mapping and AI visions.

| Stage A: Orientation Workshop—23 February 2020 to 17 March 2020 | | |
|---|---|---|
| **Theme (in Order of Priority)** | **Example** | **Sustainability-Supporting AI** |
| **Benchmarking sustainability** | e.g., baseline price for food, information campaigns on nutritional content to create value and connect the cost of food to land/animal stewardship, what is in the price, sustainable reporting frameworks, sustainable food systems benchmarks | Create new baselines and benchmarks<br>Create information for consumers<br>Connect and deepen relationships and networks |
| **Food as health** | e.g., food as medicine (nutrient contents knowledge base and indicators, tools to ascertain Brix content, link sustainability and health) joint metrics, initiatives on ingredient substitution models for seasonal/cultural/diets, soil as health (healthy soil knowledge base and indicators) | Create new baselines and benchmarks<br>Create information for consumers and producers |
| **Collaborative farming** | e.g., coops or foundations for collaborative work, farmer to farmer digital networks using mobile phones (i.e., weFarm), seed prediction and supply models, interactive/real-time connection to farms, succession planning (decision support knowledge centred, profiling one farm), systemic evaluation of crops in Victoria and their support networks | Connect and deepen relationships and networks<br>Leverage prediction and modelling capabilities to support decentralized and resilient farming<br>Create and maintain a sustainability knowledge-base<br>Create new baselines and benchmarks |
| **Research & development hub** | e.g., DIY precision-farming weeds, resilience in soil and crops, DIY disease and fertiliser recognition and application, platforms for discovery and testing of AI-powered research for farmers and by farmers, farming tool development around digital for regenerative (small/poly/perm) farming | Potential opportunities through robotics<br>Leverage detection capabilities<br>Create and maintain an AI knowledge-base<br>Develop AI for regenerative faming tools |

Stage C Scoping the initiative—15–23 June 2020: In Stage C, knowledge and consensus were built from the ground-up; nothing was off the table and assumptions were challenged with a beginner's mind [87]. When asked to qualify what the VFMA extended panel intended by the term "food quality", most of the panel selected "regenerative" out of the 28 initial words and expressions the panel proposed. The panel further qualified it with the terms "flavoursome", "delicious", or "tasty", followed closely by "nutrient-dense" and "with nourishing qualities" (Table 4 for the top four definitions).

**Table 4.** Stage C—definition of food quality for the VFMA extended panel and some thinking about measuring it. Considerations of AI are provided.

| Term Selected (in Order of Priority) | General Definition (from Participant Input) | Sustainability-Supporting AI (Conceivable Ways of Measuring or Assessing) |
|---|---|---|
| Regenerative | Regenerative agriculture is a conservation and rehabilitation approach to food and farming systems. It focuses on topsoil regeneration, increasing biodiversity, improving the water cycle, enhancing ecosystem services, supporting biosequestration, increasing resilience to climate change, and strengthening the health and vitality of farm soil. Practices include recycling as much farm waste as possible and adding composted material from sources outside the farm. General definition: renewal or restoration of a body, bodily part, or biological system (such as a forest) after injury or as a normal process. | Investigate frameworks to support a holistic view of the quality of food from the quality of the practices employed to grow the food to the health benefits of food quality. Investigate frameworks for Regen Ag where AI can bring a clear advantage or innovation (e.g., provenance, traceability, sensors in soil to nutrient quality). |
| Nutrient-dense | Food that is high in nutrients but low in calories. Nutrient-dense foods contain vitamins, minerals, complex carbohydrates, lean protein, and healthy fats. Nutrient density identifies the amount of beneficial nutrients in a food product in proportion to e.g., energy content, weight, or amount of detrimental nutrients. | Ability to link quality and sustainability of growing practices to quality intrinsic to the food produce. Frameworks exist that will need to be evaluated in the context of the project. |
| Delicious | Appealing to one of the bodily senses especially of taste or smell. Delectable, flavoursome, luscious, mouth-watering, savoury, tasty. | This would be difficult to measure because it is usually linked to personal preferences. However, if we have 1000 shoppers rating a produce as "delicious" then it would become a quantifiable measurement. |
| Fresh | Full of or renewed in vigour, not stale, sour, or decayed, not altered by processing. Excellent shelf life/storage as a consequence of nutrient density and minimal handling. | Possibility to link education around what freshness in direct supply chain means (not stored in fridges for months, not processed, locally produced, and locally accessed. Measurement of ripeness, ripeness for purpose (i.e., for raw eating, for jams, etc.). |

The exercise revealed a culture concerned about presenting a holistic view of quality, food production, and food preparation. When asked to provide samples of readings or references the panel wanted to use to define "quality", the panel's choices displayed a genuine love for food that is grown following regenerative principles, which look after the environment food is grown in, and results in food with great nutrient integrity [88] (p. 370) due to ecological health and a superior taste. The panel cared about food that is of its time

(seasonal) and of its place (terroir). This knowledge is enabled by an "ecological-literacy toolkit" shared in local communities of healthy people with access to "nutrient-dense" food [89]. The intent of the project was defined by measuring quality as a link between soil health, food nutrient density, and people's health, and can be further expressed in the words of the Australian Food Sovereignty Alliance [90] (p. 4): "As small producers, we are concerned with the following questions: how can we help support food security by creating nutrient-dense fresh food through the adoption of alternative farming practices? How can we create a system of farming that enriches the soil and environment rather than depleting it? How can we be more self-sustaining, so that our reliance on external fertility (and other inputs) is reduced?".

The panel's intent and definition of "quality" were used to guide discussions with the external panel, who were asked to provide suggestions of quality measurement, which might then be enabled by AI-powered tools. The results of the external panel's inputs relate to finding and proposing ways of measuring holistic quality attributes of the farmers' markets produce using AI.

Similarly, building the participants knowledge of AI through the codesign activities and transdisciplinary platform allowed them to integrate that knowledge in an increasingly complex manner, aligned with VFMA priorities. The design process brought sustainability and AI into the same frame of thinking. For example, in Stage A, we asked: "How might we use AI to support and enhance the sustainability of farming and food?". In Stage B, the framing evolved slightly towards proposing to "cooperatively work with VFMA to conduct a number of experiments which could use AI as a tool to address typical problems of scale encountered in their field". Stage C contributed to crystallising a view where "AI can be used to scale up information on produce quality". Stage C+ questioned the assumptions raised in Stage C around the definition of quality. Finally, Stage D was based on a hypothesis discussed with the extended panel: "How might we create an evidence-based story of quality at VFMA so that producers and consumers find value in it?".

Finally, the prioritisation table provided a framework for VFMA to reflect on the findings of the design process and the input from the different panels. VFMA decided to evaluate a holistic approach that can tell the whole story of quality. AI-tools using machine vision and machine learning were selected to conduct an experiment on quality-based relationships between varieties selection, health of the soil, quality of the produce, and potential benefits to the health of the consumers.

The intent of the project was to focus on the advancement of sustainability at the Victorian Farmers' Markets Association while experimenting with AI-powered solutions as an enabler. The collaborative work on the problem to be solved, the parallel research on AI-based tools, and a holistic approach to food quality led the panel to select an experiment that could link sustainable growing practices amongst their members to the quality of the produce. The experiment is intended to be used as a tool among the VFMA's community of members and consumers to measure and learn about nutrients, freshness, ripeness of market produce, and the effect of healthy growing practices on product quality. The tool will ultimately be able to be used on a mobile phone without any technical training required. Conducting the experiment generated a large amount of information, which will be further detailed in an upcoming article.

Furthermore, the transdisciplinary platform, codesign activities, and evaluation framework created a knowledge-base on AI for sustainability, which matured over the course of the case study and culminated in the VFMA designing their own.

### 3.2. AI Ethics in Practice

Some of the AI ethics principles selected at the start of the case study (Section 1.2) to guide the VFMA project closely align to priorities of the sustainability field: human, social, and environmental wellbeing, human-centred values, and fairness. In that regard, applying sustainability principles meant that AI ethics principles were also included. As

indicated in Section 1.3, AI ethics is mainly guided by top-down approaches, which do not provide any practice-based guidelines on how to approach the subject from a bottom-up, in the field, viewpoint.

The AI ethics principles' implementation in practice was based on 'in the moment' awareness, guided questioning, and reflection after the fact. VFMA participants carried the Association's values, ethical principles, and trust, built over the years, in all collaborative sessions. Questions of ethics were raised in the moment through reflection and evolved from general ethics towards more technology-oriented ethics as the project progressed. Stage A revealed the extended panel's concerns: the participants discussed "open technology" with very few constraints and restrictions on their use [91] to create a level-playing field for small scale farmers. During Stage C, the extended panel paid attention to customers who might "use tech without understanding the underlying drivers" and, therefore, create well-intentioned proposals where the quest for efficiency made things much worse [92]. "Data without background information" (extended panel) or working on leading-edge technology which "might alienate people" (extended panel) were part of the questions raised by the panel when populating the lean canvas (Section 2.2). For example, identifying high quality vegetables and fruits for consumers would mean that producers would not be able to sell their second-grade produce.

Questioning the project in terms of AI ethics was part of an ongoing process of evaluation brought by the reflective nature of the iterative work. This meant that data management, data sovereignty, and intellectual property were all discussed in the initial stages but also later when evaluating AI tools or designing contracts with potential solution providers. Best practice data policy informed approaches to what data the project would ask of members and customers, where it would be stored, what data would be shared, with whom it would be shared, and how transparent the tool decision-making process would be.

Finally, the process of iterating through learning cycles meant the participants had a chance to reflect on the progress of the project, a reflection-on-action facilitated through weekly review of the work. These provided opportunities to evaluate what was learned, what was missed, and what consequences might be adverse. For example, reviewing some of the AI tools the group came across, one solution provider indicated they were thinking about using pest detection capabilities to potentially offer advice from pesticides companies. This was found to be against the ethical principles of the project and was raised with the supplier in an active codesign conversation of what their solution could do to reinforce non-petrochemical based solutions.

Further on in the project, when conducting the experiment, additional dimensions of AI ethics were raised, which will be covered in a future article.

### 3.3. AI in SFS: Solution or Framework?

When the research proposal was put to the VFMA, we clarified that "the research might or might not produce AI-powered tools but will, at a minimum, create the environment for such tools to be implemented where they support the project objectives in an ethical and sustainable manner". As explained in Section 3.1, Stage A identified a broad-brush approach to framing this project as a "research and development hub" and a "platform for discovery and testing of AI-powered research" for the VFMA members. In Stage B, proposed initiatives included looking into a "sustainable diet substitution model" that would allow online customers to define parameters for sustainable diets based on science and provide suggestions of available market produce, which could meet these parameters. Another participant suggested "implementing an interactive sustainable food system platform", which would focus on sustainable food systems indicators for the selection and delivery of food from the VFMA members. A third initiative was listed as a research outcome of the project: an innovation process to rapid test AI solutions in the VFMA community to support sustainable food systems outcomes. While Stage C, C+, and D

converged towards applying AI specifically to the selected initiative and implementing it, the design process created shared knowledge among the participants.

In a manner like the integration of sustainability in the design process (Section 3.1), knowledge about AI, AI-tools, and potential applications of AI to sustainability was folded within the design activities. With the VFMA panels had little or no knowledge of AI, introduction to the technology relied on the prior knowledge and research of the author and the extended panel participants. Weeks of collaborative work and transdisciplinary engagement uncovered seventeen potential AI-powered solutions, two of which obtained a higher rating across all dimensions of the evaluation framework (Section 2.3). Here is a sample of the 17 solutions the work uncovered (see Supplemental Materials Table S1 for more details):

- Big data takes stock of climate change risks on agricultural land. This could be pushed further by keeping track of farm inputs to understand practices that reduce or limit that risk over time. It opens the way to valuation that takes sustainable practices into consideration and potentially provides higher access to funding for growers who invest in adaptation and mitigation.
- Machine vision can enable classification of pests and diseases or identify nutrient deficiency, a proactive avenue to reduce or eliminate losses. It also provides options for more efficient estimation of ripeness and quality, a better-timed harvest, which could save on growers' food losses and consumers' food waste. Other applications are related to growth prediction and pruning regimes, and some have started investigating soil analysis opportunities.
- Machine learning and handheld spectrometers are changing the approach towards the consistency of quality in cheese production to provide a proactive measurement of fat and protein content, which could then be acted upon during the process, rather than after the fact. The possibility to measure nutrient density and, therefore, a system to detect value for money would benefit producers of high quality, high nutrient, high flavour, and longer shelf-life produce.
- Technology such as blockchain can be of direct benefit to sustainable supply chains where networks of suppliers and consumers can exchange verified information on food provenance, sustainable practices, and the distribution of revenue (to ensure fair trading practices).

This is not a list of all AI applications available but rather the ones that were uncovered by the project to "tell the story of quality" (extended panel). These were evaluated by the internal panel, who commented on each solution and rated them individually (Section 2.3). The rating helped prioritise the solutions more likely to solve or advance understandings of the selected initiative. For example, when reviewing Solution #5 from an AI ethics viewpoint (Figure 7), the participants were able to comment on five out of eight sub-dimensions (Section 2.3).

| AI Ethics | | | | | | | |
|---|---|---|---|---|---|---|---|
| Human, social and environmental wellbeing | Human-centred values | Fairness | Privacy protection and security | Reliability and Safety | Transparency and explainability | Contestability | Accountability |
| *Climate change metrics included. Need to review human and social metrics to be possibly included.* | | Data provided by farmers are free to farmers. | In line with the Farmers Federation data policy. | *As a product, typical agreement on availability and reliance of systems is provided. To be further confirmed with project go ahead.* | Data sources are clearly identified. Ai used for data analytics only. No impact on human mande decision, data access only for decisions to be made outside the system. | | |

**Figure 7.** Evaluation of AI ethics principles for one of the shortlisted AI-powered tools.

Similarly, the decision to consider sustainable food systems indicators (Section 1.3) and review each solution in light of the opportunities it creates, further deepens the relationship between the solution, its impact, and the transition to sustainable food systems.

Following the results of the codesign activities covered in this case study and the prioritisation of the AI tools evaluated, an experiment was conducted using a hybrid solution based on the top two AI tools reviewed: mobile phone-microscopy device and a spectrometer). The mobile-phone microscopy device can be used to take magnified pictures of rainbow chard and cos lettuce. The device clips onto any smartphone and generates images for AI analysis. The tool labels the pictures based on an analysis of the produce soluble solids, a parameter used to determine ripeness in vegetables and fruit and used in quality assessment in the food supply chain. Details of this component of the case study are not presented here but will be the focus of a future article.

## 4. Discussion and Conclusions

### 4.1. Reflection on the Research Questions

The participants' contribution influenced the problem definition and solution selection process by eliciting convergent and divergent thinking inside the panels. The external panel tended to help the other panel participants to zoom out and use divergent thinking in the initial stages, and helped zoom in to converge on a set of solutions during the later stages. Conversations about and with AI experts were conducted during all stages of the VFMA project but more intensely when the panels collaboratively converged towards selected solutions. Further research should investigate in more details the evolution of the maturity of the sustainability and AI discourse in these conversations.

The case study answered most research questions but with some noticeable changes. To the question "Can the design process support the selection of AI tools for sustainable food systems?", the study achieved something slightly different: it integrated the ideas and possibilities of AI as part of the thinking used to scale up sustainable food systems solutions. It supported the selection of tools, but also the identification of and discussion about the tools as they were brought up by the different panel participants.

As we saw in Section 3.1, sustainability intent was carried throughout the case study by the inherent values and infrastructure in place but also by the design process. The sustainability intent ultimately defined the type of AI solution that was selected, or rather, designed. Instead of just looking at solutions, the VFMA focused on what they valued most, then redesigned existing AI tools as part of a holistic approach to making progress on these values. This is a bottom-up approach to working with AI in sustainability, driven by a collaborative creation process. It is also an avenue for small groups with little budget or technology experience who take the opportunity to consider using AI tools to scale up their sustainability efforts without engaging with Big Data, cost and energy intensive solutions. It could also be an avenue for Big Ag with considerable budgets, because technology-driven solutions do not help deal with the complexity of food systems and risk further entrenching the current unsustainable food systems paradigms.

Questions of ethics, such as the use, accessibility, security, or fairness of technology, were raised as part of the collaborative process. Most AI-specific ethics questions were guided by the design process and used at the start of the process (when defining the scope of work) and at the end of the process (when evaluating potential AI tools), or else were reflected upon later in the process, when negotiating contracts. Questions directly related to the algorithmic concerns (i.e., bias, explainability) were only considered when selecting the type of solutions that existed and will be more relevant to the experiment stage (which is outside the scope of this article). However, this is an area that should be mainly undertaken with computing engineers and would require they adhere to rules and laws that are not yet defined.

Key strategies to avoid technical solutionism rely on multi-disciplinary engagement of diverse groups of people [8,93] and healthy scepticism of universalist and solutionist notions of design [69]. The design activities in this case study provided a frame to pursue

both sustainability [94] and AI ethics [47] as a process, two areas that take into consideration the complexity of the questions at hand and the need to progress as we learn. The project has since concluded, but it is not finished; the VFMA are ready to further investigate all that was uncovered. The process has created a level of understanding and knowledge among the participants that has made them comfortable about thinking about AI in their pursuit of sustainability to the extent that they now see the work as a "backlog of potential initiatives" (internal panel) that they will draw upon in the months and years to come. Additionally, the codesign engagement model, agile iterative approach, and experiment-based learning of the project has provided a model for rapid testing of AI at the VFMA. This opens the discussion to further research on the considerations of AI for food sustainability as a solution or rather, as in this case study, as a framework. The tools can be used to support the creation of an integral and holistic view of food that is grown using sustainable practices, diverse and nutrient-rich, a support to its community of customers but also to their health. In that context, AI-powered tools could enable the creation of knowledge, increased and closer relationships, and better-informed customers who can reward producers with established practices that are respectful of their environment. More scrutiny about the environmental impacts of the technology is still required to ensure they are appropriately accounted for in sustainability terms. While research of AI for sustainability (towards the UN's sustainable development goals) is the subject of numerous articles, questions about the sustainability of AI (through the reduction of its carbon emissions and computing power) are still hidden in the development process [95].

### 4.2. A Process to Feed the Engine of Transition

To anyone who has baked a flourless cake or an angel cake and learned to fold in egg whites, the concept of iterative integration will be quite familiar. The batter is made of ingredients as varied as melted chocolate, almond meal, flour, and sugar. Mixed haphazardly, these ingredients can result in a thick paste. To bring a more open texture and end up with a light and delicious cake, one needs to beat egg whites into a foam and gradually fold them in with the other ingredients. One must first add just one spoonful of the egg whites to the batter mixture. Once it is mixed, an additional third of the batter is folded in using a delicate flipping motion to literally fold the egg whites into the batter. The batter becomes light and the last third of the egg whites can then be incorporated, folding in further with great care to keep the air inside the batter and, ensure a light cake texture. The two textures need to bind together to create a result that is better than the sum of its parts. If one tries to incorporate the two textures in one motion, the lightness and airiness of the egg whites dissolve into a liquid and the recipe fails.

The above analogy conveys the importance of an iterative integration of practices and knowledge creation observed in this case study. Including AI in the scope of the project, the approach has been one of gradual introduction of the subject at levels relevant to what the participants knew. This created a background knowledge about AI in food systems among the participants and provided them with a familiarity and degree of confidence about what could be trialled and what would fit their priorities and capabilities. The creation of knowledge happened in iteration through the codesign activities, regular weekly reviews, and monthly sharing of progress and guidance.

Iterations and codesign allowed the gradual folding in of knowledge and expertise. Each iteration built on the previous one and opened the space (aerating the batter) where the various levels and types of knowledge could be shared (binding the batter) in a way that frees up innovation with tools the participants had never used before, creating a result bigger than the sum of the parts. Allowing for a progressive build-up of knowledge suits working with AI because it is difficult for participants who have never worked with it to imagine what it is and what they could achieve with it.

Based on this analysis, Figure 8 illustrates how the codesign and iterative processes integrated the domains of design intent, strategic intent, and transdisciplinary expertise in the case study.

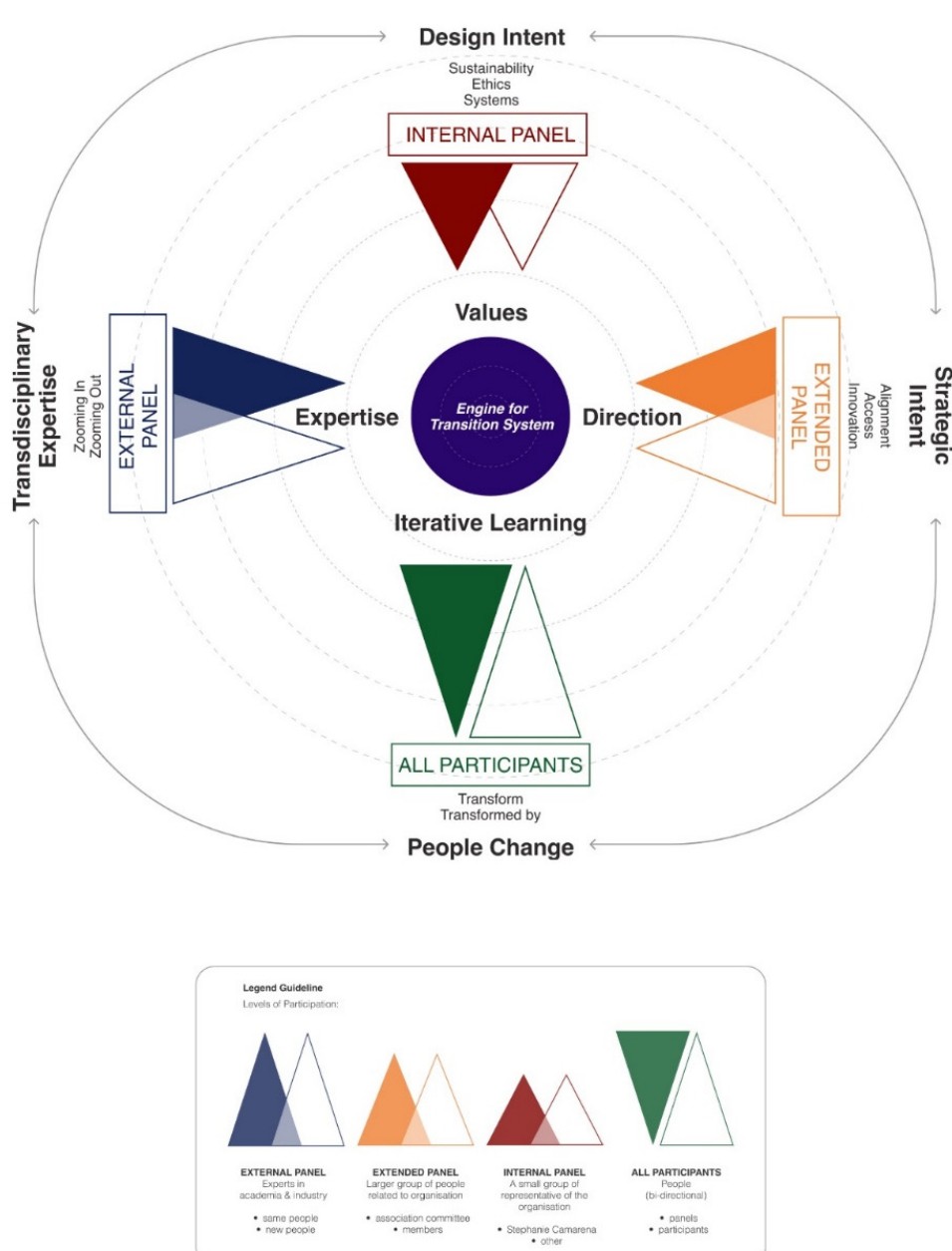

**Figure 8.** Embedding levels of input and direction through codesign activities and iterative practices.

The design intent was carried by the internal panel. Their role was to guide the activities within the focus defined at the start of the project: sustainability, ethics, systems thinking. Strategic intent was carried by the extended panel whose role was to ensure alignment to the association's strategic goals, interest, and benefit of the members. Finally, transdisciplinary expertise was carried by the external panel in collaboration with the other panels. Their role was more indirect in that they were not involved in codesign activities, but their input helped shape, define, and align them. The fourth domain was centred around learning. This gave preference to codesigning in iterations where the discovery phase is permanently ongoing and the dialogue is always open [65] (p. 31)

rather than linear (from problem to solution). This format acknowledges that solutions impact people; they are changed by it. Therefore, planning set times for sharing and reflecting on what was learned is important to the participatory process and to evaluating the progress towards the set intent. Understanding the different facets of the initiative and the solutions uncovered by 'doing' in multi-disciplinary settings informed the constant realignment of the solution to acknowledge what was learned. For example, some of the initial assumptions about regenerative agriculture or nutrient density were refined over time, influenced by a wide range of experts in the field and the pragmatic requirements of defining ways of measuring it.

Through the work done in this case study and the subsequent experiments, the VFMA community, along with participating experts, have created a body of knowledge and a method to experiment ideas quickly and scale-up sustainability efforts. They identified AI solutions and created an experiment that is still anchored in people's practices and sustainable priorities. The process has removed any hesitation to investigate AI as part of an approach to deepen and share the sustainability knowledge and practice created by and for the community.

The case study's findings propose that when considering AI for the transition to sustainable food systems, the following process should be considered:

- Engage with AI with a definite intent to achieve sustainable, ethical, and systemic outcomes (if it is not the main aim, then it is only going to be an after-thought)
- Codesign in partnership with the organisations' participants with respect for the strategic intent they bring and the diversity they represent
- Codesign with experts to help divergent thinking when needed (zooming out) and to consolidate options (zooming in) when required
- Use iterative methods to create reflective practice, validated learning, and a re-assessment of activities and outcomes against the values set
- Allow people to be changed by the process and to change the solution in turn
- Focus on outcomes that privilege deep leverage points (Section 1.3)

This process is a powerful antidote to technological solutionism, particularly when the question "we can but should we?" is revisited at regular intervals. Furthermore, by creating a shared knowledge of AI and its impact (positive or negative) on sustainability and ethics, we, as a community, take responsibility for what we create rather than having technology 'done to us'. This will help develop the muscles we need for social-technical-environmental balance. The questions of design innovation, AI ethics, sustainable food systems, and systems leverage points should be scrutinised when looking at using AI in transition to sustainable food systems. As such, the research offers potential opportunities to shift paradigms by proposing a model of what should be scrutinised, providing questions on what is supposed to be asked and probed, and proposing how to structure the questions of how the results should be interpreted, how experiments should be conducted, and what equipment to use [96].

**Supplementary Materials:** The following are available online at https://www.mdpi.com/article/10.3390/su13169314/s1, Figure S1 is the strategy blueprint which was developed during the project Stage B and used in Stage C. Table S1 is a de-identified list of the 17 AI-powered solutions discovered then evaluated in Stage D. Requests for additional data will be made available upon reasonable request.

**Funding:** Supported through the Australian Government Research Training Program Scholarship. This project was partially supported by funding from Food Agility CRC Ltd., funded under the Commonwealth Government CRC Program. The CRC Program supports industry-led collaborations between industry, researchers and the community.

**Institutional Review Board Statement:** This study was approved by RMIT University Human Research Ethics Committee—Approval # 21722 dated the 12 October 2018 and subsequent amendments.

**Informed Consent Statement:** In line with the RMIT University human research ethics approval, informed consent was obtained from all subjects involved in the study.

**Acknowledgments:** The author would like to thank the amazing participants who so generously contributed to this project.

**Conflicts of Interest:** The author declares no conflict of interest.

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
