# Peer review of "Engaging with Artificial Intelligence (AI) with a Bottom-Up Approach for the Purpose of Sustainability: Victorian Farmers Market Association, Melbourne Australia"

_sustainability, doi:10.3390/su13169314_

Round 1

Reviewer 1 Report

The paper is well written in present form.

Author Response

Re: Manuscript sustainability-1291675: Engaging with artificial intelligence (AI) from the bottom-up for the purpose of sustainability: Victorian Farmers Market Association, Melbourne Australia

Dear Reviewer,

I refer to your review of the paper titled “Engaging with artificial intelligence (AI) from the bottom-up for the purpose of sustainability: Victorian Farmers Market Association, Melbourne Australia”, Manuscript sustainability-1291675.

I wish to thank you for your valuable comments.

We have undertaken a major revision of the paper, based on the reviewers’ comments. The author considers the revised paper is appropriate for publication in the Sustainability journal.

We thank you for your acceptance of this paper for publication.

Kind regards,

Stephanie

Reviewer

Comment #

Comment

Author Reponse

Reviewer 1

01-01

This paper is well written in its current form.

We thank the reviewer for taking the time to review this paper for accepting it in its current form.

Reviewer 2 Report

This article builds on a Victorian Farmers' Markets Association (VFMA) study that explored the complexities of using AI tools to increase sustainability and resilience for the benefit of organizations and members.

There is a clearly defined scientific goal and a unique research aspect.

It is not clear the effectiveness of the research and the thesis formulated depends to a large extent on the selection of criteria, as well as the quantity, scope and quality of data. What is the universality of the solution? Does it have any limitations? I propose to describe in more detail on what basis and what results from the scope of data selection on which the research was based.

The article requires minor text corrections.

Author Response

Re: Manuscript sustainability-1291675: Engaging with artificial intelligence (AI) from the bottom-up for the purpose of sustainability: Victorian Farmers Market Association, Melbourne Australia

Dear Reviewer,

I refer to your review of the paper titled “Engaging with artificial intelligence (AI) from the bottom-up for the purpose of sustainability: Victorian Farmers Market Association, Melbourne Australia”, Manuscript sustainability-1291675.

I wish to thank you for your valuable comments.

We have undertaken a major revision of the paper, based on the reviewers’ comments. The author considers the revised paper is more appropriate for publication in the Sustainability journal.

We look forward to hearing your acceptance of this paper for publication.

Kind Regards,

Stephanie

Reviewer

Comment #

Comment

Author Reponse

Reviewer 2

02-01

This article builds on a Victorian Farmers' Markets Association (VFMA) study that explored the complexities of using AI tools to increase sustainability and resilience for the benefit of organizations and members.

There is a clearly defined scientific goal and a unique research aspect.

We thank the reviewer for taking the time to review this paper and for this comment.

02-02

It is not clear the effectiveness of the research and the thesis formulated depends to a large extent on the selection of criteria, as well as the quantity, scope and quality of data. What is the universality of the solution? Does it have any limitations? I propose to describe in more detail on what basis and what results from the scope of data selection on which the research was based.

Significant changes have been made to the paper to more strongly demonstrate the universality of the solution and the contribution to knowledge. This includes modifications to the introduction, bringing more clarity to the methodology and adding further data in the results and discussion sections. Additional tables and figures were added to the argument and supplemental material. Furthermore, additional considerations about the limitations of the study have also been added. We hope that this will answer the reviewer's concern on these different areas.

Reviewer 3 Report

The under review article overviews the findings which derived from a study of the Victorian Farmers’ Markets Association (VFMA) investigating the complexity of using AI tools to enhance sustainability and resilience for the benefit of the organisation and its members.

As it appears from studying the article, although its subject could be utterly interesting in general, the degree of sufficiently addressing to the readership of the journal seems to be questionable with regard to adding to knowledge. Additionally, the novelty of reported method as well as its benefits towards sustainability is not quite adequately justified.

In particular:

  • More in-depth information on the issue should be presented in order to justify the research motivations and substantiate the research question as well as the hypotheses concerning the importance of introducing this study. The subject of the study focuses on the of How does the result of the study could be further expanded to other organizations and what are its benefits? This issue should be the concern of the author. However no hints on this important issue is included in the paper...
  • Background information regarding Artificial Intelligence technologies in agriculture should be given for the benefit of the readership. Additionally, more information should be given regarding the sustainability issue as well as the related policies.
  • The structure of the paper should be outlined at the end of the introduction section.
  • Section 2 is advised to be renamed. The title “Methodology and Methods” could be characterised as redundant. Moreover, the way the methodology is presented is rather chaotic and does not assist the reader to understand the point of this study.
  • The deriving results should be introduced more clearly and be supported with an adequate number of properly illustrated tables and charts. The author is advised to give attention in this issue in order to substantiate the presented results with regard to adding to knowledge.
  • The author is suggested to emphasize more in the conclusions deriving from the study and justify their added value as well as its benefits toward the research community.
  • The supplementary material could be included as a figure in the main manuscript.
  • Finally, the author should probably reconsider the title of the article to be changed in order to be more to the point. For instance what does it mean “Engaging with artificial intelligence (AI) from the bottom-up…” ?
  • The paper is written in appropriate and understandable English language, yet some minor spellchecking might be needed.

Conclusively the author is suggested to further work on the presentation of the research and strongly substantiate its novelty in order to resubmit a new improved manuscript.

Author Response

Re: Manuscript sustainability-1291675: Engaging with artificial intelligence (AI) from the bottom-up for the purpose of sustainability: Victorian Farmers Market Association, Melbourne Australia

Dear Reviewer,

I refer to your review of the paper titled “Engaging with artificial intelligence (AI) from the bottom-up for the purpose of sustainability: Victorian Farmers Market Association, Melbourne Australia”, Manuscript sustainability-1291675.

I wish to thank you for your valuable comments.

We have undertaken a major revision of the paper, based on the reviewers’ comments. The author considers the revised paper is more appropriate for publication in the Sustainability journal.

We look forward to hearing your acceptance of this paper for publication.

Kind Regards,

Stephanie

Reviewer

Comment #

Comment

Author Reponse

Reviewer 3

03-01

As it appears from studying the article, although its subject could be utterly interesting in general, the degree of sufficiently addressing to the readership of the journal seems to be questionable with regard to adding to knowledge. Additionally, the novelty of reported method as well as its benefits towards sustainability is not quite adequately justified.

We thank the reviewer for taking the time to review this paper and for this comment.

We agree with the comment and have made significant changes to address this. In particular, we have responded individually to the following comments and hope that this answers the reviewers' concerns both on addressing the readership of the journal and on the benefits of the method.

As it appears from studying the article, although its subject could be utterly interesting in general, the degree of sufficiently addressing to the readership of the journal seems to be questionable with regard to adding to knowledge. Additionally, the novelty of reported method as well as its benefits towards sustainability is not quite adequately justified.

03-02

More in-depth information on the issue should be presented in order to justify the research motivations and substantiate the research question as well as the hypotheses concerning the importance of introducing this study.

The introduction and background have been modified to bring more clarity to the motivations for the study. We hope that it brings the existing content into a sharper contrast.

03-03

The subject of the study focuses on the of How does the result of the study could be further expanded to other organizations and what are its benefits? This issue should be the concern of the author. However no hints on this important issue is included in the paper...

We thank the reviewer for this comment. Additional material has ben added to the methods and results sections to provide stronger evidence of how the study could be replicated in other organisations.

The results section has been modified to include additional tables and figures to strengthen the case for the universality of the proposed approach.

03-04

Background information regarding Artificial Intelligence technologies in agriculture should be given for the benefit of the readership.

We agree with the reviewer that background information should be added on this subject and have done so in the background section 1.2 and in the discussion.

03-05

Additionally, more information should be given regarding the sustainability issue as well as the related policies.

More information was added to indicate how the sustainability background and policies in the background section 1.2 and 1.3.

03-06

The structure of the paper should be outlined at the end of the introduction section.

The structure of the paper is available at the end of the introduction, below the background and directly above the following Methods section.

03-07

Section 2 is advised to be renamed. The title “Methodology and Methods” could be characterised as redundant.

Section 2 named has been changed to Methodology to remove the redundancy.

03-08

Moreover, the way the methodology is presented is rather chaotic and does not assist the reader to understand the point of this study

We thank the reviewer for this comment.

The methodology section has been edited by changing the title of the subsections which we hope brings more clarity to what they represent. 

Additional elements have been added in the introduction to the section and inside the subsections to clarify the argument and better assist the reader. We hope that this answers the reviewer's concerns.

03-09

The deriving results should be introduced more clearly and be supported with an adequate number of properly illustrated tables and charts.

Additional information has been provided in the introduction to the section, including the insertion of the supplementary material as advised by the reviewer.

Further information was added to the subsections which we hope substantiate the results more adequately. This includes the modification of some the content into a table format with additional information displayed, an additional table to illustrate results of Stage C+.

The author is advised to give attention in this issue in order to substantiate the presented results with regard to adding to knowledge.

As per above, we believe the additional material will further substantiate the contribution to knowledge.

Additional tables and figures were added to clarify and extend the results.

The author is suggested to emphasize more in the conclusions deriving from the study and justify their added value as well as its benefits toward the research community.

The discussion and conclusion section was edited to put the results into context and provide further evidence of the benefits to the research community.

The supplementary material could be included as a figure in the main manuscript.

The project chart was added in the introduction to the results section.

Additionally, new supplementary material has been added which we believe will further clarify the benefits of the proposed approach to the research community.

Finally, the author should probably reconsider the title of the article to be changed in order to be more to the point. For instance what does it mean “Engaging with artificial intelligence (AI) from the bottom-up…” ?

We thank the reviewer for this comment.

The title has been modified to more closely match the specifics of the study. Additionally, references to the explanation of a bottom-up approach has been included in the manuscript to more clearly link the research and the title.

The paper is written in appropriate and understandable English language, yet some minor spellchecking might be needed.

We thank the reviewer for this comment and confirm that the manuscript has been further reviewed for minor spell-checking and made the corrections.

03-10

Conclusively the author is suggested to further work on the presentation of the research and strongly substantiate its novelty in order to resubmit a new improved manuscript.

We thank the reviewer and hope that this revised manuscript will present the research more clearly and more strongly demonstrate its novelty.

Round 2

Reviewer 2 Report

I accept this revised article.

Reviewer 3 Report

Dear author,

You have made a great effort to answer my queries and comments, and, obviously, the result of the manuscript's review is very positive.

I believe that in its current form the manuscript is extremely improved and that it achieves the conditions for publication.

I do not have to add any comments to the revised version.

Thank you